# A multi-ethnic polygenic risk score is associated with hypertension prevalence and progression throughout adulthood

Nuzulul Kurniansyah[1], Matthew O. Goodman[1,2], Tanika N. Kelly[3], Tali Elfassy [4], Kerri L. Wiggins [5], Joshua C. Bis [5], Xiuqing Guo [6], Walter Palmas[7], Kent D. Taylor[6], Henry J. Lin [6], Jeffrey Haessler[8], Yan Gao[9], Daichi Shimbo[10], Jennifer A. Smith [11], Bing Yu[12], Elena V. Feofanova [12], Roelof A. J. Smit[13], Zhe Wang [13], Shih-Jen Hwang[14], Simin Liu [15], Sylvia Wassertheil-Smoller [16], JoAnn E. Manson[2,17], Donald M. Lloyd-Jones[18], Stephen S. Rich [19], Ruth J. F. Loos [13], Susan Redline [1,2], Adolfo Correa [20], Charles Kooperberg [8], Myriam Fornage [12,21], Robert C. Kaplan[8,22], Bruce M. Psaty [23], Jerome I. Rotter [6], Donna K. Arnett [24], Alanna C. Morrison [12], Nora Franceschini[25], Daniel Levy[26,27], the NHLBI Trans-Omics in Precision Medicine (TOPMed) Consortium* & Tamar Sofer [1,2,28 ✉]

In a multi-stage analysis of 52,436 individuals aged 17-90 across diverse cohorts and biobanks, we train, test, and evaluate a polygenic risk score (PRS) for hypertension risk and progression. The PRS is trained using genome-wide association studies (GWAS) for systolic, diastolic blood pressure, and hypertension, respectively. For each trait, PRS is selected by optimizing the coefficient of variation (CV) across estimated effect sizes from multiple potential PRS using the same GWAS, after which the 3 trait-specific PRSs are combined via an unweighted sum called "PRSsum", forming the HTN-PRS. The HTN-PRS is associated with both prevalent and incident hypertension at 4-6 years of follow up. This association is further confirmed in age-stratified analysis. In an independent biobank of 40,201 individuals, the HTN-PRS is confirmed to be predictive of increased risk for coronary artery disease, ischemic stroke, type 2 diabetes, and chronic kidney disease.

A full list of author affiliations appears at the end of the paper.

Hypertension affects over 1.1 billion people in the world[1]. Globally, the number of people with hypertension has increased over time, reflecting the aging of the population and is predicted to reach 1.56 billion people by 2025[2]. Hypertension is a leading risk factor for cardiovascular, kidney, cerebrovascular disease, and a leading cause of global mortality[3–5]. The causes of hypertension are genetic and environmental, including dietary factors, and the rising prevalence of obesity[6–8]. Genome-wide association studies (GWAS) have identified more than 900 genomic regions associated with blood pressure (BP) phenotypes[9–14], and GWAS from diverse race/ethnic backgrounds as well as admixture mapping studies demonstrate that BP phenotypes have some ancestry-specific or ancestry-enriched genetic components (e.g. genetic variants that are more common in one continental genetic ancestry)[15–20].

Polygenic Risk Scores (PRS) estimate the effect of many genetic variants on an individual's genetic susceptibility to a disease or trait, typically calculated as a weighted sum of trait-associated alleles, with weights often being the effect estimates corresponding to each allele. PRS are typically constructed using results from GWAS to guide the selection of single nucleotide polymorphisms (SNPs) into the PRS, and their weights[21,22]. Developing PRS that are useful across a diverse, globally representative population remains a challenge when the underlying GWAS is performed primarily in people of European ancestries[23–25]. A recent study of PRS for hypertension found that a BP PRS was useful in predicting longitudinal development of hypertension in a Finnish population[26]. With the availability of recent, large multi-ethnic and non-European GWAS of BP phenotypes, such as from the Million Veteran Program (MVP), the UK Biobank (UKBB), and Biobank Japan (BBJ)[10,27], it is possible to include SNPs that are common in genetic ancestries that are traditionally less represented in GWAS, permitting the construction of multi-ethnic PRS for hypertension risk prediction[28].

Here we leverage a multi-ethnic dataset, with harmonized genotypes and phenotypes, from the Trans-Omics in Precision Medicine Initiative (TOPMed) program[29,30] to construct and assess PRS for hypertension based on summary statistics from multiple GWAS of hypertension and BP phenotypes. Individuals were from a few U.S. race/ethnic backgrounds: African Americans (AA, also referred to as Black individuals), Hispanic/Latino Americans (HA), Asian Americans (AsA), and European Americans (EA, also referred to as White individuals), allowing for assessment of the PRS across major U.S. demographic segments. Our use of two names for the same race/ethnic background group reflects the fact that these are socially constructed groups and preferred identifications vary by TOPMed studies and their participants. We used multiple independent subsets of the TOPMed dataset to train, test, and assess PRS associations with hypertension across the lifespan. We evaluated the association of the final HTN-PRS with incident outcomes in the Mass General Brigham (MGB) Biobank. To develop the PRS, we propose a new approach for selecting tuning parameters for PRS construction, based on optimizing the coefficient of variation of the effect size estimates of five independent subsets of the training dataset, as well as combination of PRS based on GWAS of multiple BP phenotypes into a single PRS.

## Results

Figure 1 provides an overview of the study. At stage 1, we used summary statistics from multiple GWAS of BP phenotypes to construct PRS in a training dataset (stage 1 dataset) with prevalent hypertension. We selected GWAS that were based on individuals not overlapping with our dataset (published Million Veterans Program GWAS[10], and summary statistics from the UK Biobank database). We used a clump & threshold methodology which requires selection of tuning parameters. Importantly, we evaluated a few methods to select tuning parameters, and an approach to combine PRS across phenotypes. At stage 2, we further assessed the methods for tuning parameter selection and the combined PRS in a stage 2 dataset using prevalent hypertension at a baseline exam. We selected the best performing PRS, and used it in analyses of PRS association with hypertension using data from two visits in the stage 2 dataset. At stage 3, we studied the PRS association with incident hypertension in young Black and White adults, using a longitudinal, stage 3 dataset with 6 exams. At stage 4, we tested the association of the PRS with disease status in individuals from the MGB Biobank (stage 4 dataset).

Supplementary Table S2 characterizes the stage 1 dataset, used for training the PRS using prevalent hypertension analysis. Rates of hypertension among the race/ethnic groups ranged from 56% with 32% treated (AsA) to 79% with 57% treated (AA). Mean age ranged from 53 (AsA) to 58 (HA). Supplementary Table S3 characterizes the sample across the eight studies participating in the stage 2 dataset. The data were collected over two time-points with an average of 4-6 years between measures. There were 39,035 individuals in the analytic sample, of which 22,701 were EA, 8822 were AA, 6718 were HA, and 794 were AsA. The characteristics of the race/ethnic background groups were quite heterogeneous. The average age across backgrounds ranged from 51 (HA) to 62 (AsA) at baseline. The EA group had the highest proportion of female participants (72.2%) while the HA group had the lowest (62%). The number of hypertension cases increased between the exams in all background groups. AAs had the highest proportion of hypertension cases in both exams: 76.5% and 53% treated (baseline), and 83.3% and 66% treated (follow-up). HAs had the fewest cases: with 51.7% and 23% treated (baseline) and 59.6% and 39.5% treated (follow-up).

**PRS tuning parameters selection based on stage 1 dataset**. Based on each GWAS, we selected PRS using three criteria: Genome-wide Significant PRS; Selected CV-PRS; Selected PVAL-PRS. Supplementary Fig. 1 describes the association of each PRS with hypertension in the stage 1 dataset. Selected CV-PRS had the highest AUC compared to other PRS. Supplementary Fig. 2 further reports these associations for the secondary GWAS as well, showing that they mostly performed less well than the primary GWAS, with the exception of the PRS based on the UKBB + ICBP EA GWAS[9], in which all TOPMed EA individuals participated. The meta-analysis of all available independent GWAS performed less well than the primary GWAS. Supplementary Table S4 reports the clumping parameters and number of SNPs for each of the primary and secondary GWAS and each selection criterion.

**PRS associations with baseline hypertension in the stage 2 dataset**. Figure 2 demonstrated the trained PRS associations with prevalent hypertension at baseline in the stage 2 dataset. PRS were associated with prevalent hypertension and showed a similar association patterns as in the training dataset, with the exception that here Selected CV-PRS often having higher OR and AUC for each PRS. This pattern was more pronounced for SBP and PRSsum. Here, Selected CV-PRS often had lower $p$ value compared to the Selected PVAL-PRS. PRSsum based on selected CV-PRS had the strongest association with hypertension (OR = 2.10, 95% CI [1.99, 2.21], $p$ value $<1 \times 10^{-100}$, AUC = 0.76). Based on these results, we move forward with PRSsum based on Selected CV-PRS for analysis of incident hypertension. Figure 3 shows the distributions of Selected CV-PRS based on each GWAS and their

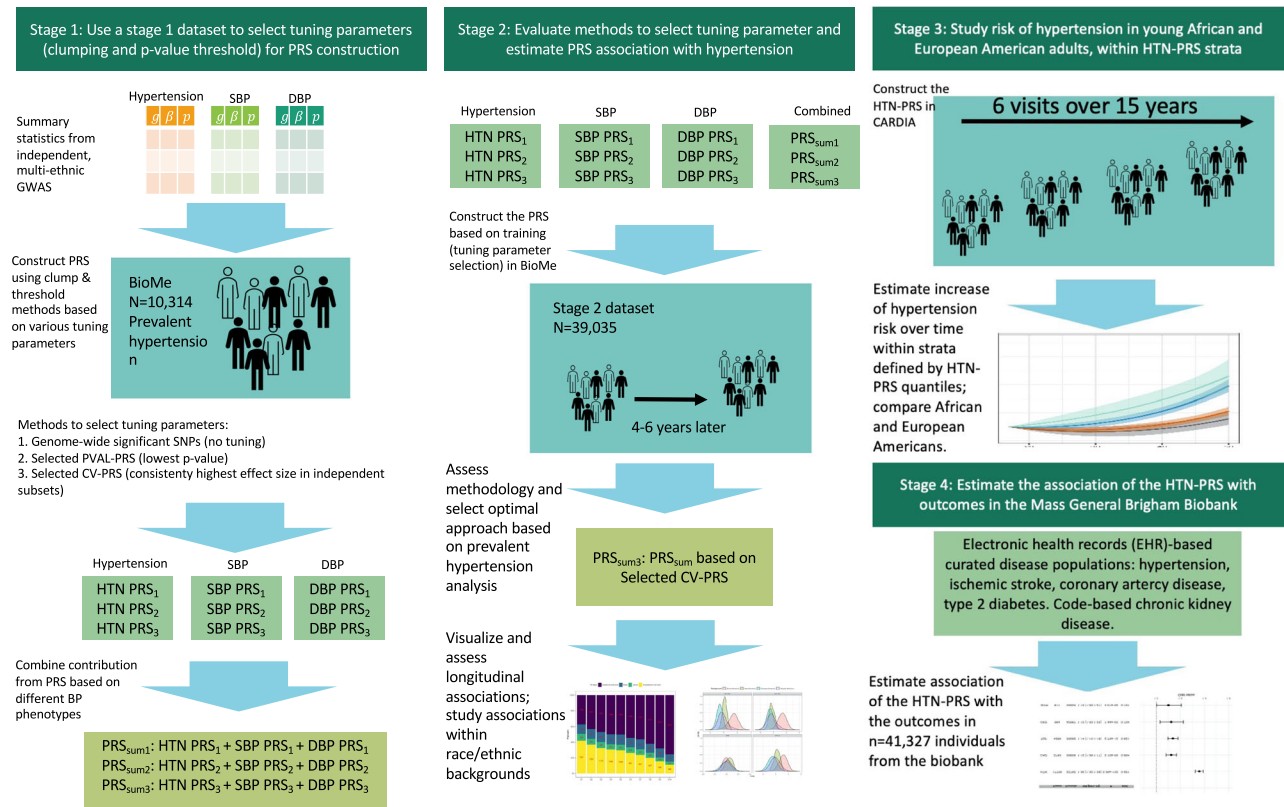

**Fig. 1 Study organization.** In stage 1, we used the stage 1 dataset to select tuning parameters for PRS construction based on GWAS of BP phenotypes. We compared a few methods for tuning parameter selection and constructed PRSsum combining a few phenotype-specific PRS. In stage 2 we evaluate the methods for tuning parameter selection in the stage 2 dataset, and selected one PRS, namely HTN-PRS, to move forward for two-visit longitudinal analysis. In stage 3, we used a longitudinal dataset from CARDIA to study hypertension development in young adults, and compared Black and White individuals. In stage 4, we tested the association of the HTN-PRS with disease outcomes in MGB biobank (stage 4 dataset).

combined PRSsum. For all PRS, the AA group tended to have the highest PRS values. We computed the correlation between PRS and stratified by race/ethnic background as described in Supplementary Fig. 5. As expected, PRSsum based on Selected CV showed a strong correlation with each PRS. In what follows, we refer to PRSsum based on Selected CV-PRS as the HTN-PRS, for brevity.

In secondary analysis, we compared the HTN-PRS to four additional PRS constructed using approaches that specifically model pleiotropy between the BP traits. Results are provided in Supplementary Fig. 12 (stage 1 dataset) and 13 (stage 2 dataset) in the Supplementary Information. The HTN-PRS had better performance.

**Distributions of longitudinal BP categories across deciles of the HNT-PRS.** Figure 4 visualizes the distribution of the longitudinal BP categories across deciles of the HTN-PRS, and Supplementary Table S5 in the Supplementary Information provides results from an analysis using linear regression to test for a linear change in the number of individuals from each BP category as a function of HTN-PRS decile. Indeed, higher deciles have higher proportions of individuals with severe BP category (having hypertension already at baseline) with $p$ value < 0.001 indicating increase in the number of individuals in this category in each decile, and lower deciles have higher proportions of individuals who were free of hypertension in both exams ($p$ value < 0.001 indicating a decrease in the number of individuals in the "always normal or elevated" category with increasing HTN-PRS deciles). Relatively few individuals were categorized as "worsened" or "improved" (transitioning between normal BP, elevated BP, and

HTN between exams). No association was observed with the number of individuals in the "worsened" category in HTN-PRS deciles ($p$ value = 0.21), while the number of individuals in the "improved" category decreased with increasing HTN-PRS deciles ($p$ value < 0.001). Supplementary Fig. 3 shows similar data stratified by race/ethnic background and demonstrates generally similar patterns across backgrounds, except for AsA, who are also the group of the smallest sample size ($n = 794$), and therefore there is higher uncertainty in results for this group. Supplementary Fig. 4 visualizes similar data stratified by age decades at baseline (≤20, 21–30, 31–40, … 71–80, >80). We observed longitudinal associations of the HTN-PRS with BP category is most age groups (age 31 to age 80), with most the pronounced associations from ages 41–70, for which each severity category is well represented in the data.

**HTN-PRS association with prevalent and incident hypertension across race/ethnic backgrounds.** Figure 5 demonstrates the association of the HTN-PRS with three hypertension measures: prevalent hypertension at baseline, new onset hypertension among individuals with normal BP at baseline, and new onset hypertension among individuals with elevated BP at baseline. In the multi-ethnic analysis, the PRS was associated with each of the measures, with strongest association with hypertension at baseline (OR = 2.10, 95% CI [1.99, 2.21], $p$ value < $1 \times 10^{-100}$, AUC = 0.76), while new onset hypertension among individuals with normal BP at baseline had OR = 1.72, 95% CI [1.55, 1.91], $p$ value = $4.67 \times 10^{-24}$, AUC = 0.66, and among those with elevated BP at baseline had OR = 1.48, 95% CI [1.27, 1.71], $p$ value = $2.39 \times 10^{-7}$, AUC = 0.59. Stratifying the association by

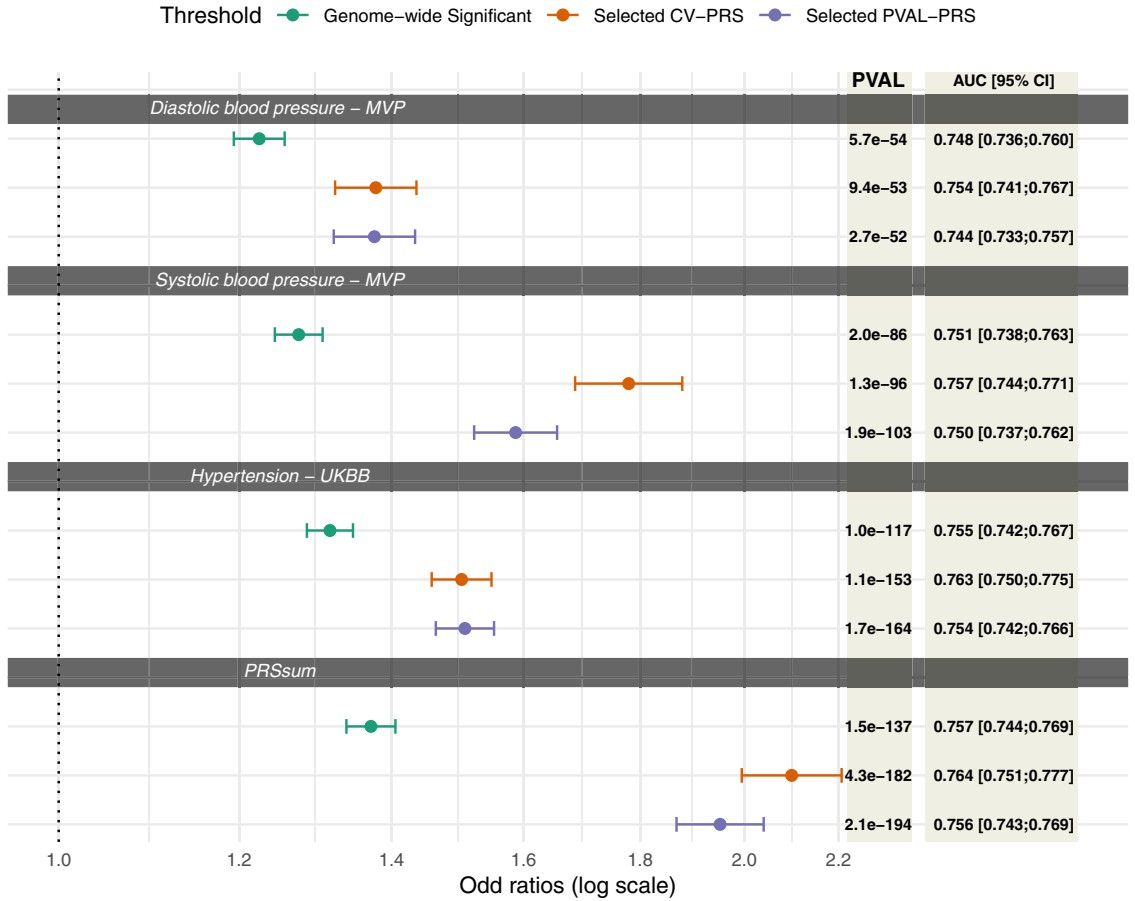

**Fig. 2 Association of PRS with prevalent hypertension at baseline in the stage 2 data set.** Associations of PRS in stage 2 dataset ($N = 37,667$ individuals). PRS were trained for hypertension association using stage 1 dataset. "Genome-wide significant PRS" are PRS constructed using genome-wide significant SNPs in the discovery GWAS, with fixed LD parameters or $R^2 = 0.1$ and distance = 1000 kb. "Selected CV PRS" are PRS that minimized the coefficient of variation (CV) across effect size (log odds ratio (OR)) estimates in 5 independent subsets of the stage 1 dataset. "Selected PVAL-PRS" are PRS that minimized the association $p$ value with hypertension in the stage 1 dataset. Each point provides the estimated OR per 1 standard deviation (SD) increase of the PRS, and error bars represent 95% confidence intervals (CIs). For each PRS association the figure also provides the $p$ value of the estimated association with hypertension based on the Wald test (chi-squared test with one degree of freedom based on two-sided alternative hypothesis), and Area Under the Receiver Operator Curve (AUC). PRS associations were estimated in models adjusted for sex, age, $age^2$, study site, race/ethnic background, smoking status, BMI, and 11 ancestral principal components. PRS SDs were defined according to the sampling SDs of the PRS estimated in the entire TOPMed dataset.

race/ethnic background and testing for heterogeneity suggested differences in the PRS association with hypertension at baseline (heterogeneity $p$ value $< 1.0 \times 10^{-4}$) but weak evidence for heterogeneity otherwise, perhaps due to decreased sample sizes and lower power. Overall, the PRS had the weakest estimated effect sizes, for all outcomes, in the AA group. Supplementary Fig. 6 reports an analysis mimicking that in Fig. 5, with now effect sizes reported per 1 SD increase in the PRS where the SD is computed within the group, rather than in all TOPMed. Within European and Asian Americans, the ORs per SD become lower when accounting for group-specific PRS distribution. In secondary analysis, we computed the risk of hypertension at baseline in top versus bottom decile of the PRS within each race/ethnic background. Results are provided in Supplementary Fig. 7. The association was strongest in the HA group (OR = 4.33, 95% CI [2.81, 6.68]) followed by the EA, AsA, and AA groups.

Supplementary Fig. 8 further provides results from association analyses stratified by age decade at baseline. Significant ($p < 0.05$) associations with prevalent and with new onset hypertension are observed throughout adulthood, starting with the 21–30 age group, and up to the 71–80 age group, with the exception that association of incident hypertension among individuals with elevated BP in the

21–40 and 51–60 age groups were not statistically significant. This could be due to low sample sizes (see figure for more details).

Supplementary Fig. 9 provides a comparison of effect sizes and predictive performance measured by AUC of the multi-ethnic PRS, BMI, and current smoking, in prevalent and incident hypertension analyses. The PRS is comparable to BMI, and both perform better than current smoking.

**PRS Association with development of hypertension in young Black and White adults.** We estimated trajectories of hypertension development as second order polynomial functions of age within strata defined by quantiles of the PRS. Characteristics of the stage 4 CARDIA dataset are provided in Supplementary Table 6. As shown in Fig. 5, in the combined analysis of both Black and White CARDIA participants, trajectories of hypertension risk are separated between the PRS-defined strata, with individuals in strata defined by higher PRS values consistently develop higher hypertension risk compared to those in lower PRS strata. Note that Black individuals obtain higher PRS values compared to White individuals and vice versa. Looking at the race-defined strata, this pattern is seen much more clearly in Black individuals, but less so in White individuals

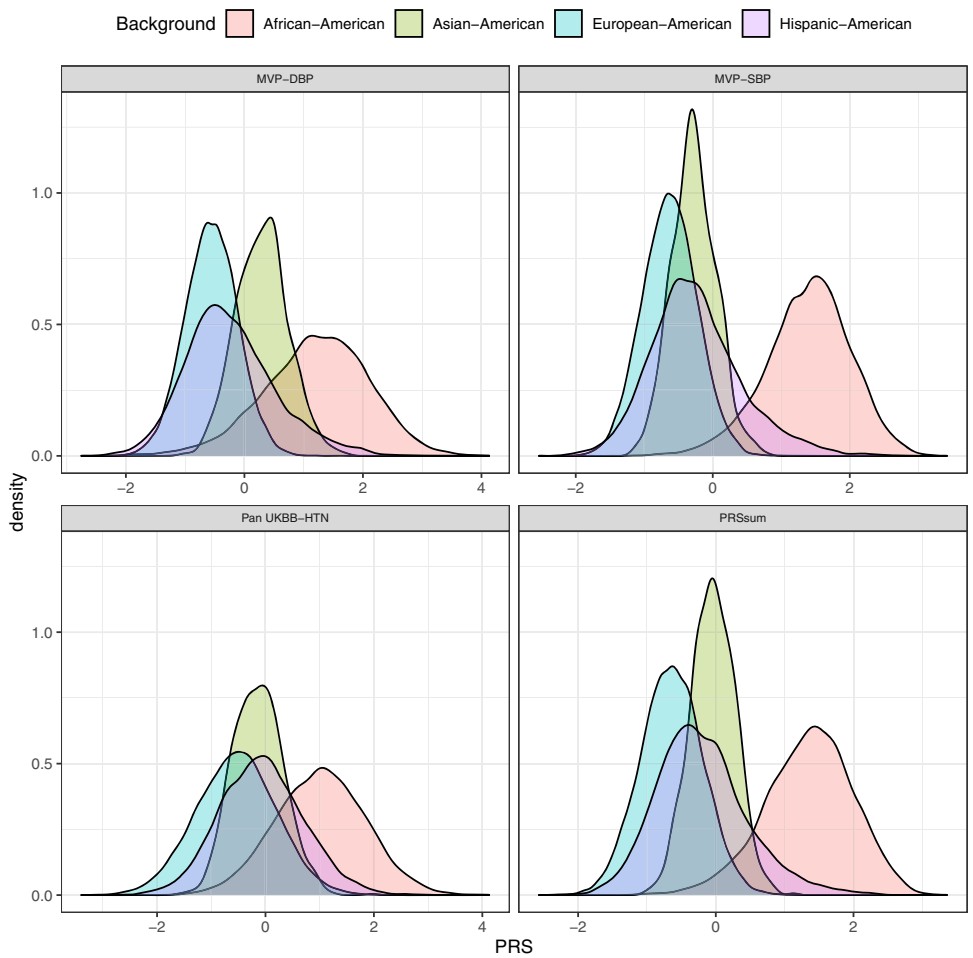

**Fig. 3 PRS distribution stratified by race/ethnic background.** Density plots showing the distributions of Selected CV-PRS based on each GWAS used (Table 1) and PRSsum constructed by summing Selected CV-PRS from the three GWAS (the final HTN-PRS). The figure was created using the stage 2 dataset. The densities are stratified by race/ethnic background.

alone, suggesting that Black individuals with high PRS values (compared to other Black individuals) are likely to develop hypertension earlier than White individuals with high PRS values (compared to other White individuals). At age 50, Black individuals at the 90–100% percentile of the HTN-PRS had 2.11 OR (95% CI [1.77, 2.50]) for the risk of hypertension compared that in age 17, individuals at the 50–90% stratum had 2.07 OR (95% CI [1.90, 2.23]) relative to age 17, and individuals at the 10–50% stratum had 1.76 OR (95% CI [1.63, 1.91]) relative to age 17. Individuals at the bottom stratum, 0–10%, had 1.43 the OR (95% CI [1.23, 1.67]). In contrast, in White individuals, the ORs at age 50 relative to age 17 are 1.43, 1.27, 1.22, 1.15 for the 90–100%, 50–90%, 10–50%, and 1–10% strata, respectively.

**PRS association with disease outcomes in MGB Biobank.** Supplementary Fig. 10 describes the association of the multiethnic hypertension PRS with hypertension, ischemic stroke, CAD, type 2 diabetes, CKD, and obesity, in the MGB Biobank. In multi-ethnic analysis, the PRS was associated with hypertension (OR = 1.45, $p$ value < $9 \times 10^{-100}$), as well as with all other outcomes (OR = 1.1–1.4 for all outcomes, $p$ value < 0.05). Association of the HTN-PRS with obesity is likely because the panancestry UKBB GWAS of hypertension, which we used, was not adjusted for BMI, and/or due to residual effects of BMI that were not fully accounted for by the BMI adjustment in the MVP GWAS.

**Secondary analysis of the HTN-PRS using genetic ancestry**. In an additional secondary analysis, we created groups of individuals defined by having at least 80% of a specific genetic ancestry: at baseline, 5447 individuals with at least 80% African ancestry, and similarly 97, 783, and 20,939 individuals with at least 80% Amerindian, East Asian, and European ancestry. Sample sizes are smaller when excluding individuals with hypertension at baseline. Notably, most of the HA individuals could not participate in this analysis because they do not have a single predominant ancestry. The distributions of the PRS in each of these groups (Supplementary Fig. 14) show that PRS distributions differ between ancestries, due to differences in allele frequencies between them. Supplementary Fig. 15 further provides results from association analysis with hypertension at baseline and incident hypertension in the stage 2 dataset. While the effect size of the PRS per 1 SD of the PRS (with SD computed over all the TOPMed dataset) is largest in the European ancestry group, at the baseline hypertension analysis, the AUC is about the same in the African (0.76) and European ancestry (0.75) groups.

**Discussion**
We developed a HTN-PRS based on multi-ethnic GWAS for SBP, DBP and assessed its association in a multi-ethnic TOPMed dataset. A BioMe stage 1 dataset was used to select optimal tuning parameters for PRS based on each GWAS using three approaches: the novel Selected CV-PRS approach, Selected PVAL-PRS, and

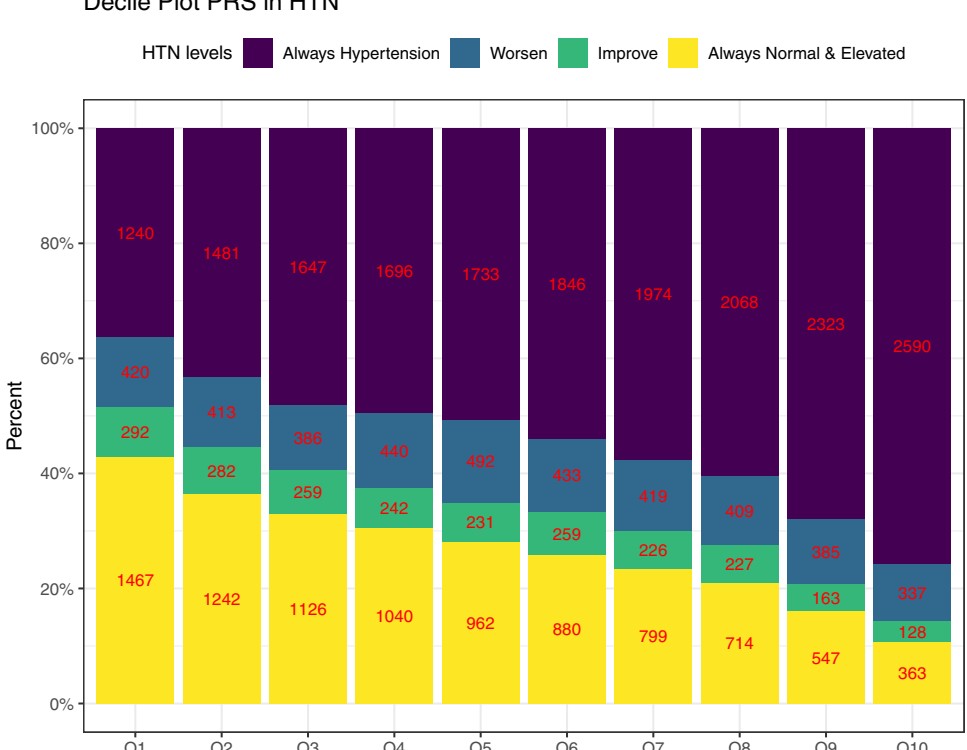

**Fig. 4 Distribution of longitudinal categories of BP by deciles of the HTN-PRS.** The figure visualizes the distribution of longitudinal BP categories in the stage 2 dataset: hypertension at both exams (treated and/or having hypertension in both exams), worsen (individuals having worse BP category in the follow-up exam compared to the first) improved (individuals having better BP category in the second exam compared to the first, only if they were not treated for hypertension at any point), and no hypertension in both exams (includes normal and elevated BP but without change in category), in deciles of the multi-ethnic HTN-PRS. The numbers provide the sample sizes represented by each bar.

genome-wide significant PRS. We further proposed to combine BP phenotypes PRS based on GWAS of different phenotypes using the PRSsum approach: an unweighted sum of the separate phenotypes' PRS. This final HTN-PRS, PRSsum based on Selected CV-PRS, was associated with hypertension prevalence in the independent stage 2 dataset, as well as with longitudinal categories of BP trajectories across race/ethnic backgrounds. In analysis stratified by age decade, the association of the PRS with both prevalent and incident hypertension is consistent across ages 21–80. In the stage 3 CARDIA study of young adults with 15-years follow up, individuals in strata defined by the top decile of the PRS developed hypertension earlier, especially Black individuals, who tend to have higher PRS values compared to White individuals. Thus, the HTN-PRS can be potentially useful for assessing risk for developing hypertension throughout adulthood. Finally, the HTN-PRS was significantly associated with cardiovascular outcomes in the MGB Biobank (stage 4 dataset).

Recently, a study in Finnish Europeans from FinnGen[26], studied the use of BP PRS to predict longitudinal and lifetime risk of hypertension. The PRS were highly associated hypertension and with cardiovascular disease (CVD) risk, underscoring the potential of PRS to predict hypertension and stratify individuals for intervention to potentially reduce CVD risk. Here, we addressed a similar problem while focusing on a multi-ethnic population and on 4–6 years from between two exams. The distribution of the various constructed PRS, including the final HTN-PRS (PRSsum based on Selected CV-PRS), differed across race/ethnic backgrounds. This is expected, because PRS are sums of alleles, which have different distributions (defined by allele frequencies) across genetic ancestries, and therefore, also race/ethnic background, as these generally have different ancestry

admixture. Indeed, PRS distributions also differed when assessed over groups constructed using individuals with high proportions of specific genetic ancestries. While we expect PRS values in the upper decile to be associated with higher risk of hypertension across all race/ethnicities, a natural question is how to define individuals as "at risk". An "at risk" classification may use a specific cut-off value of the PRS, which may be based on a percentile of the distribution[31]. Clearly, this approach cannot be used when distributions differ across race/ethnicities, and moreover, admixed individuals are not accurately represented by any specific distribution. Therefore, more work is needed on approaches that do not require categorization of neither individuals nor of specific PRS values to define risk. Rather, models that take into account multiple risk factors (such as demographic, clinical, and other risk factors, as well as PRS[32]) and allow for flexible association model may be more powerful and equitable, in that they could be applied to more individuals. Notably, we generally avoided using the standard approach of quantifying hypertension risk between individuals in the top HTN-PRS decile to the bottom, in the multi-ethnic analysis: such an approach would separate AAs from others (as in Fig. 6), and therefore will be confounded by other social race/ethnic-related environmental exposures that lead to increased hypertension in AAs.

It is notable that AA individuals had higher HTN-PRS values on average, compared to other backgrounds. While, as noted, these distributional differences stem from allele frequency differences, there are two questions that may be asked: First, is there any reason for these difference, such as population-level selection pressure, or are they random? Second, do these differences drive higher rates of hypertension in AA individuals? While our work cannot inform the answer to the first question, the analysis in

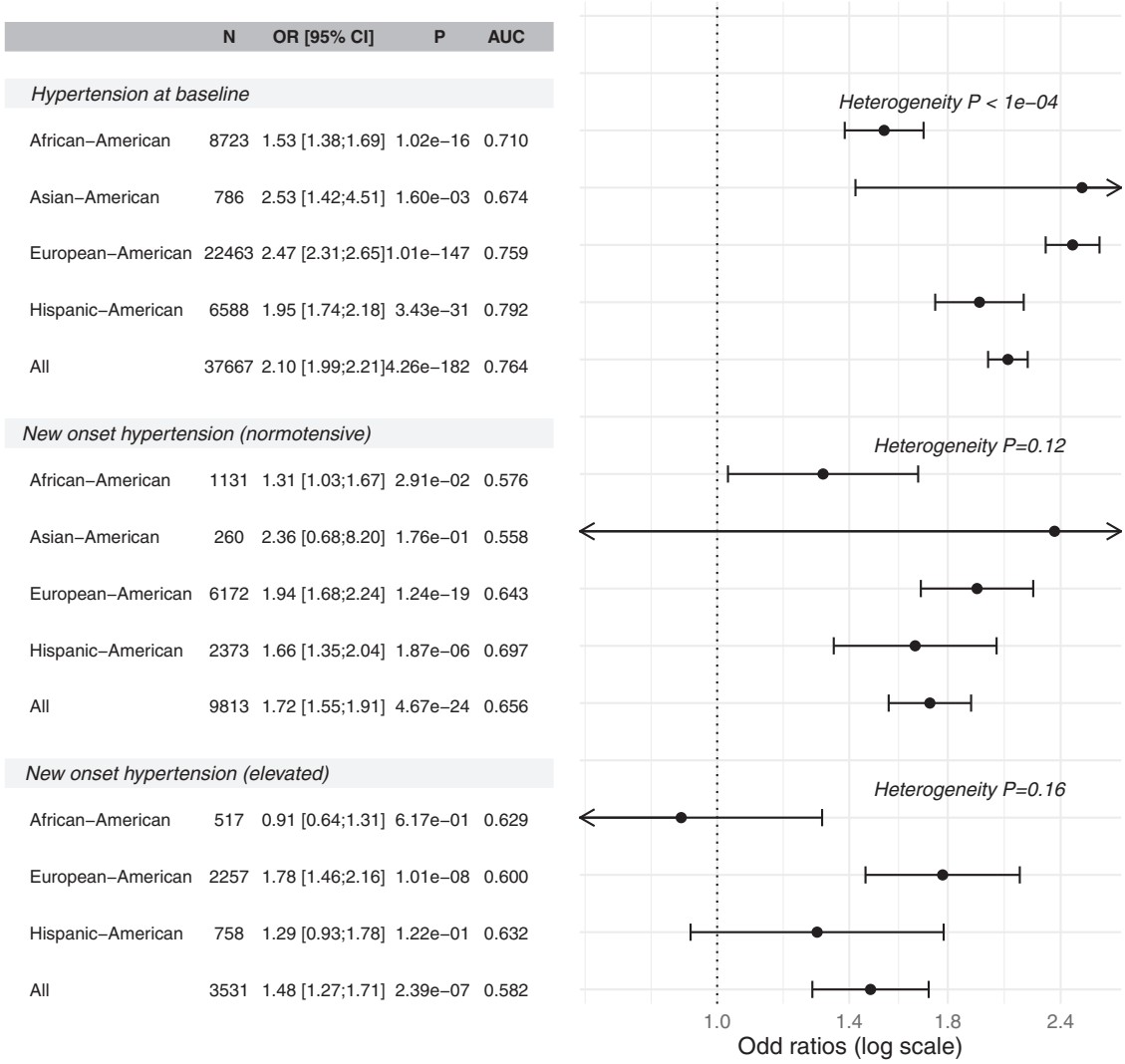

**Fig. 5 Association of HTN-PRS with hypertension measures across race/ethnicities.** The forest plot provides the association of the HTN- PRS with prevalent and incident hypertension in the stage 2 dataset, and within race/ethnic backgrounds. The top part corresponds to prevalence analysis at the baseline visit, the middle part corresponds to prediction of new onset hypertension in exam 2, among individuals who had normal BP at baseline, and the bottom part corresponds to prediction of new onset hypertension in exam 2, among individuals who had elevated BP at baseline. For each analysis the figure provides sample size (N), estimated odds ratio (OR) per 1 standard deviation (SD) increase of the PRS, and 95% confidence interval, *p* value of the association from the Wald test, and area under the receiver operating curve (AUC). Estimated ORs and confidence intervals (CIs) are provided both in the text (left part) and as points and error bars. Heterogeneity of estimated effects across race/ethnic groups was tested using the Cochran's Q test accounting for correlation due to genetic relatedness across groups. The PRS association was estimated in a model adjusted for sex, age, age[2], study site, race/ethnic background, smoking status, BMI, and 11 ancestral principal components. PRS SD was defined according to the sampling SD of the PRS estimated in the entire TOPMed dataset. Statistical tests relied on the chi-squared distribution with either one degree of freedom (for effect size estimates) or 4 or 3 degrees of freedom (when testing heterogeneity across 5 or 4 strata of race/ethnic background) based on two-sided alternative hypothesis.

CARDIA (Fig. 6) does suggest that higher PRS values are associated with earlier hypertension, supporting a genetic role for earlier hypertension in AAs. However, this should be interpreted with caution. BP has been consistently shown to be affected by lifestyle, including, lifestyle interactions with genetic variants to increase their estimated effect on BP. It is plausible that detected BP variants tend to be those that interact with dietary and other lifestyle exposures that are more common in individuals with lower socioeconomic status, such as Black Americans. In this case there could be a dual bias: bias of higher likelihood to detect specific variants that interact with such exposures, and a bias of some background groups, here AA individuals, having higher rates of the same exposures. Therefore, we cannot separate genetic factors from the dietary, lifestyle and environmental

factors as their interactions are the ultimate driver determining group differences in hypertension rates.

Methodologically, while we first constructed various PRS using a standard clump and threshold methodology[22], we used two novel approaches to construct the new HTN-PRS. First, we leveraged stage 1 and stage 2 independent datasets to study how to select the tuning parameters for a PRS, and chose the Selected CV-PRS as a method. This approach attempts to avoid overfitting to a particular dataset by splitting the training data to 5 independent subsets, i.e., with no related individuals between them, and assessing association of the PRS with the outcome in each. The Selected CV-PRS is the one that has consistent, high, effect size across these subsets, represented by smallest CV across them. Other measures can be used rather than effect size, but we

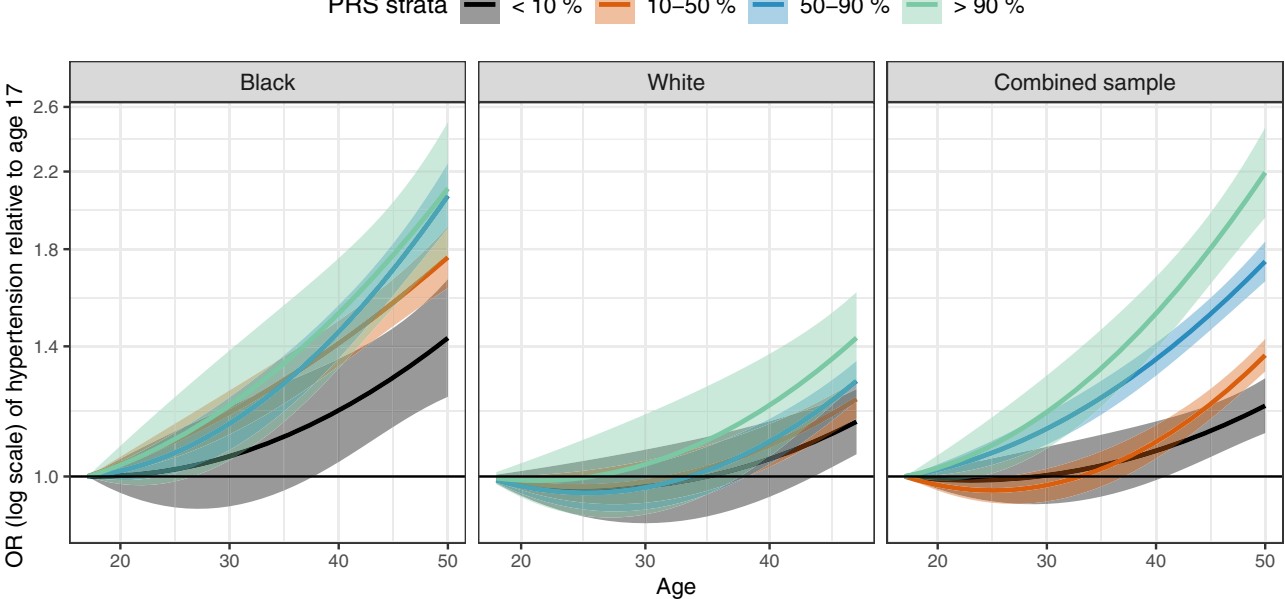

**Fig. 6 Trajectories of hypertension risk by strata defined by HTN-PRS in young adults from CARDIA.** Results from analysis of age-dependent risk of hypertension in young adults from CARDIA (stage 3 dataset). We used generalized linear mixed model to model the risk of hypertension by age within strata defined by quantiles of the HTN-PRS. Analyses were adjusted for age, sex and the first 11 PCs of genetic data. Stratification by PRS quantiles was performed in each presented group: All (combined Black and White participants), Black, and White. The effect of age was modeled using a second degree polynomial. At each point on the curve we provide 95% confidence interval of the effect estimate. In the combined sample, all individuals in the top PRS strata are Black, and 99 % of the individuals in the bottom PRS strata are White.

chose the latter because of clinical interpretability. In the testing dataset, the effect size was indeed the highest when using the Selected CV-PRS compared with other PRS. A second methodological choice was the construction of PRSsum. PRSsum allowed us to combine information across PRS that were based on GWAS of different phenotypes (SBP, DBP, hypertension). Another motivation behind it is that high values of PRSsum may capture individuals with either high SBP, DBP, or hypertension PRS values, or combined, meaning that their hypertension may be captured by various underlying genetic components. While PRSsum is an unweighted sum of PRS, a weighted sum (or with adaptive weights) can be constructed as well[33,34]. In particular, there are published methods for leveraging pleiotropy between multiple traits for both discovery of genetic associations and for creating potentially more powerful PRS for each of the traits. We attempted to implement a few of these methods, based on two models of multivariate associations between traits that account for pleiotropy[35,36]. However, PRSsum without adaptive weights had stronger associations with hypertension in both stage 1 and stage 2 datasets. We think that PRSsum was more robust because the other methods had to rely on LD inference (for estimating heritability and genetic correlations from summary statistics[37], for generating LD scores, and for genetic SBLUPs[38]). For LD inference we used our own TOPMed datasets, because we do not have access to the multi-ethnic data that were used for generating the GWAS summary statistics, and moreover, the datasets used to generate the SBP and DBP GWAS are different than the dataset used for generating the hypertension GWAS. Future work should study weighted combination of PRS in diverse populations, including where the PRS were developed based on GWAS in diverse populations.

Strengths of our study are the use of large multi-ethnic datasets with harmonized genetic and phenotypic data, a range of ages of participants, and longitudinal datasets, allowing us to explore the association of hypertension PRS across adulthood. Due to the

lower effect size of the HTN-PRS in AAs in stage 2 analysis, which included middle-aged and older adults, in stage 3 we focused on one study, CARDIA, with longer follow up of younger individuals. We compared trajectories of hypertension risk by age across Black and White individuals, demonstrating that Black individuals with high PRS values develop hypertension earlier than those with low PRS values, supporting the usefulness of the HTN-PRS in prediction hypertension across race/ethnicities. Additional longitudinal datasets from underrepresented populations are needed to study long-term trajectories of disease development and usefulness of PRS across the lifespan, especially considering sociocultural and environmental exposures that may confound association analyses due to association with hypertension, coupled with correlation with genetic ancestry. Our study also has additional weaknesses. For example, our primary analysis did not use the largest available GWAS of SBP and DBP to construct PRS, namely a meta-analysis of the European ancestry participants UKBB and of the international consortium for BP[9] as most of our study individuals participated in it, and the overlap could lead to overfitting. More work is needed to assess overfitting effects across samples sizes and overlaps of discovery GWAS, training, and testing datasets. While the GWAS that we used (MVP and Pan-UKBB) are the largest ones available with multi-ethnic populations, individuals of European ancestry individuals are still over-represented in these GWAS: 61.9% in MVP and 93% in Pan-UKBB. Future work should utilize additional, diverse, sources of summary statistics as they become available. Also, to construct PRS we used the clump & threshold methodology, rather than a more modern approach such as LDpred[39] or lassosum[40]. We chose to focus on clump & threshold methodology because we think that these other methods still need to be separately studied for diverse populations.

In summary, we applied novel methodology for developing PRS and constructed a PRS predictive of incident hypertension across adulthood in a multi-ethnic population. The PRS was also

significantly associated with clinical outcomes. Future work will incorporate rare variants and pleiotropic variants[41] in the construction of PRS, and will investigate models for clinical uses of hypertension PRS in diverse populations.

## Methods

**The TOPMed dataset.** The TOPMed dataset included 52,436 individuals from 10 TOPMed studies. Based on the characteristics of these studies, the TOPMed data was split into stage 1, stage 2, and stage 3 datasets. Stage 1 dataset was the Mount Sinai BioMe Biobank, which included 10,314 diverse individuals with prevalent hypertension status. It was used as a training dataset for constructing the hypertension PRS. Stage 2 dataset included 39,035 individuals from an additional eight studies, with all individuals being genetically unrelated (at the third degree) to those in the training dataset. Stage 2 dataset was longitudinal, with hypertension status assessed in two exams, on average 4-6 years apart. Individuals were self-reported from four predominant U.S. race/ethnic backgrounds, with 22,701 EA, 8822 AA, 6718 HA, and 794 AsA individuals. Stage 3 dataset included the CARDIA study, with 6 exams over 15 years follow up of young Black and White individuals, and was used to compare the development of hypertension risk within PRS strata across the two race/ethnic backgrounds.

**Prevalent and longitudinal measures of hypertension.** Systolic BP (SBP) and diastolic BP (DBP) were measured in each study according to methods provided in the study descriptions in Supplementary Note 3. Hypertension stages were defined according to (1) Normal BP: SBP $\leq 120$ mmHg and DBP $\leq 80$ mmHg and untreated; (2) Elevated BP: SBP between 120–129 and DBP $\leq 80$ mmHg, and untreated; (3) Hypertension: SBP $\geq 130$ mmHg, DBP $\geq 80$ mmHg, self-reported physician diagnosed hypertension, or use of anti-hypertensive medications[42]. When examining 2-exams longitudinal patterns of hypertension using the stage 2 dataset, we performed data visualization in which we categorized individuals as: not having hypertension across the two exams (healthy longitudinal trajectory; may include individuals with normal and with elevated BP, but without change in these categories between the exams); having hypertension in the two exams (severe longitudinal trajectory); BP category worsen between exams, including individuals who had normal BP at the baseline exam, and elevated BP or hypertension at the follow-up exam, or elevated BP followed by hypertension; BP category improved between exams, including individuals who were not treated by antihypertensive medications in any of the exams, and had improved BP category (hypertensive to elevated or normal, or elevated to normal). Individuals treated with anti-hypertensive medication in either baseline or follow-up exam were never categorized as "improved". We also performed association analysis of the PRS with new onset hypertension at the follow-up exam, focusing, separately, on individuals who had normal BP at baseline and who had elevated BP at baseline.

**Whole-genome sequencing.** We used whole-genome sequencing data from the Trans-Omics in Precision Medicine (TOPMed) program Freeze 8 release[29]. Only variants with minor allele frequency (MAF) $\geq 0.01$ were used in this analysis. Information about genome sequencing, variant calling, and quality control procedures is available here https://www.nhlbiwgs.org/topmed-whole-genome-sequencing-methods-freeze-8. The TOPMed Data Coordinating Center constructed a sparse kinship matrix estimating recent genetic relatedness where values were set to zero when the genetic relationship was estimated to be more distant than 4th degree relatedness, and principal components (PCs), using the PC-Relate algorithm[43].

**Published GWAS of BP phenotype.** Table 1 provides information about hypertension and BP GWAS used to construct PRS. In primary analysis, we used multi-ethnic GWAS: hypertension "pan ancestry" GWAS from UKBB (https://pan.ukbb.broadinstitute.org/), and systolic BP (SBP), and diastolic BP (DBP) from MVP[10]. Note that UKBB pan ancestry GWAS are multi-ethnic, however, U.S. minorities are not well represented compared to MVP, and therefore we prioritized MVP as a primary GWAS for SBP and DBP. All these GWAS are based on large sample sizes

and have no overlap in participants with the TOPMed BP dataset. In secondary analysis, we also used hypertension GWAS from FinnGen (https://www.finngen.fi/en) and SBP and DBP GWAS from UKBB pan ancestry and BBJ[44], and performed inverse-variance fixed-effects meta-analyses using METAL[45] for each BP trait GWAS (SBP, DBP and hypertension). These are described in Supplementary Table 1. Secondary analyses were only performed on the training dataset.

**Quality control on summary statistics.** We filtered SNPs with MAF < 0.01 in the discovery GWAS from Table 1 and/or in the dataset comprising all TOPMed analysis participants (stage 1, 2 and 3 datasets combined), and SNPs that did not pass TOPMed quality filters. We re-coded the variant positions and alleles to match those in the TOPMed data (via the UCSC hg19 to hg38 chain file) using rtracklayer R package version 1.46.0[46].

**PRS construction based on a single GWAS.** We constructed PRS using the clump-and-threshold method implemented in the PRSice 2 software version v2.3.3[22] using each of the GWAS in Table 1. Three tuning parameters are required: $p$ value threshold, and two clumping parameters. As $p$ value thresholds, we used $5 \times 10^{-8}$, $1 \times 10^{-7}$, $1 \times 10^{-6}$, $1 \times 10^{-5}$, $1 \times 10^{-4}$, $1 \times 10^{-3}$, $1 \times 10^{-2}$, 0.1, 0.2, 0.3, 0.4, 0.5. For clumping, we used the entire TOPMed dataset (stage 1, 2, and 3 datasets combined) as a reference panel for Linkage Disequilibrium (LD) and set clumping parameters to $R^2 = 0.1$, 0.2 and 0.3 and distance 250 kb, 500 kb, and 1000 Kb. To standardize PRS while keeping effect sizes comparable in all analyses, we computed the mean and standard deviations (SDs) of each type of PRS based on the complete TOPMed dataset. Then, we used these means and SDs in all subsequent analyses: for a given PRS in any dataset, we subtracted its pre-computed mean and divided it by its pre-computed SD. This standardization approach allowed for obtaining comparable effect sizes across stage 1, 2, 3, and 4 datasets, as well as across background-specific and multi-ethnic analyses. Standardization does not influence $p$ values or prediction measures.

**Using stage 1 dataset to select of tuning parameters for PRS construction.** To select tuning parameters for a PRS based on a given GWAS from Table 1 (or based on meta-analysis of multiple GWAS), we examined the association of various constructed PRS with prevalent hypertension in the stage 1 dataset. We developed the selected CV-PRS approach, which we describe below, and compare it to two additional widely-used approaches: genome-wide significance PRS, selected PVAL-PRS. Both the selected CV-PRS and the selected PVAL-PRS approaches attempt to select one set of tuning parameters ($p$ value threshold and LD clumping parameters) to construct PRS out of all tuning parameter combination used. The selected CV-PRS approach aims to identify the tuning parameters that yield consistently high PRS effect size in new, independent, datasets. To do this, it minimized the coefficient of variation (CV) computed on the PRS effect size estimates obtained from 5 equal-sized independent subsets of BioMe (this is conceptually visualized in Fig. 7). Specifically, the CV was estimated as the standard deviation of the five effect (log odds ratio) estimates, divided by the mean of these effect estimates. The selected PVAL-PRS is the PRS with the lowest association $p$ value in the stage 1 dataset. The genome-wide significance PRS was constructed using SNPs with $p$ value $< 5 \times 10^{-8}$, and fixed clumping parameters: $R^2 = 0.1$ and distance of 1000 kb, and otherwise no selection of other parameters.

**PRS construction based on multiple GWAS.** We constructed a PRS called "PRSsum" by summing PRS constructed based on the three BP phenotypes (SBP, DBP, hypertension). The three PRS were summed after each was scaled using the mean and SD computed using the entire TOPMed dataset. We summed non-adaptively, i.e., unweighted sum with PRSsum = PRS1 + PRS2 + PRS3. We generated three PRSsum, based on the three potential strategies to construct PRS: 1) PRSsum based on genome-wide significance (which summed genome-wide significant PRS); 2) PRSsum based on PVAL-PRS (which summed the selected PVAL-PRS across the three phenotypes), and 3) PRSsum based on selected CV-PRS (which summed the CV-PRS from the three phenotypes). The final multi-ethnic HTN-PRS was the one based on the approach that consistently performs better on

## Table 1 External GWAS used for hypertension PRS construction.

| GWAS name | Reference | Trait | Sample size | Population |
|---|---|---|---|---|
| MVP | PMID:30578418[10] | SBP; DBP | 318,492 (SBP); 318,891 (DBP) | Multi-ethnic (69.1% non-Hispanic White, 18.8% non-Hispanic Black, 6.7% Hispanic, 0.77% non-Hispanic Asian and 0.85% non-Hispanic Native American individuals) |
| Pan UKBB | No manuscript, downloaded from https://pan.ukbb.broadinstitute.org | HTN | 451,894 (126,196 cases, 325,698 controls) | African (1.46%), Admix American (0.21%), European (93.05%), Central/South Asian (1.96%), Middle Eastern (0.35%), and East Asian (0.60%) individuals |

The table provides GWAS source, study population as reported by the manuscript or repository reporting the GWAS, and number of participants used to generate summary statistics.

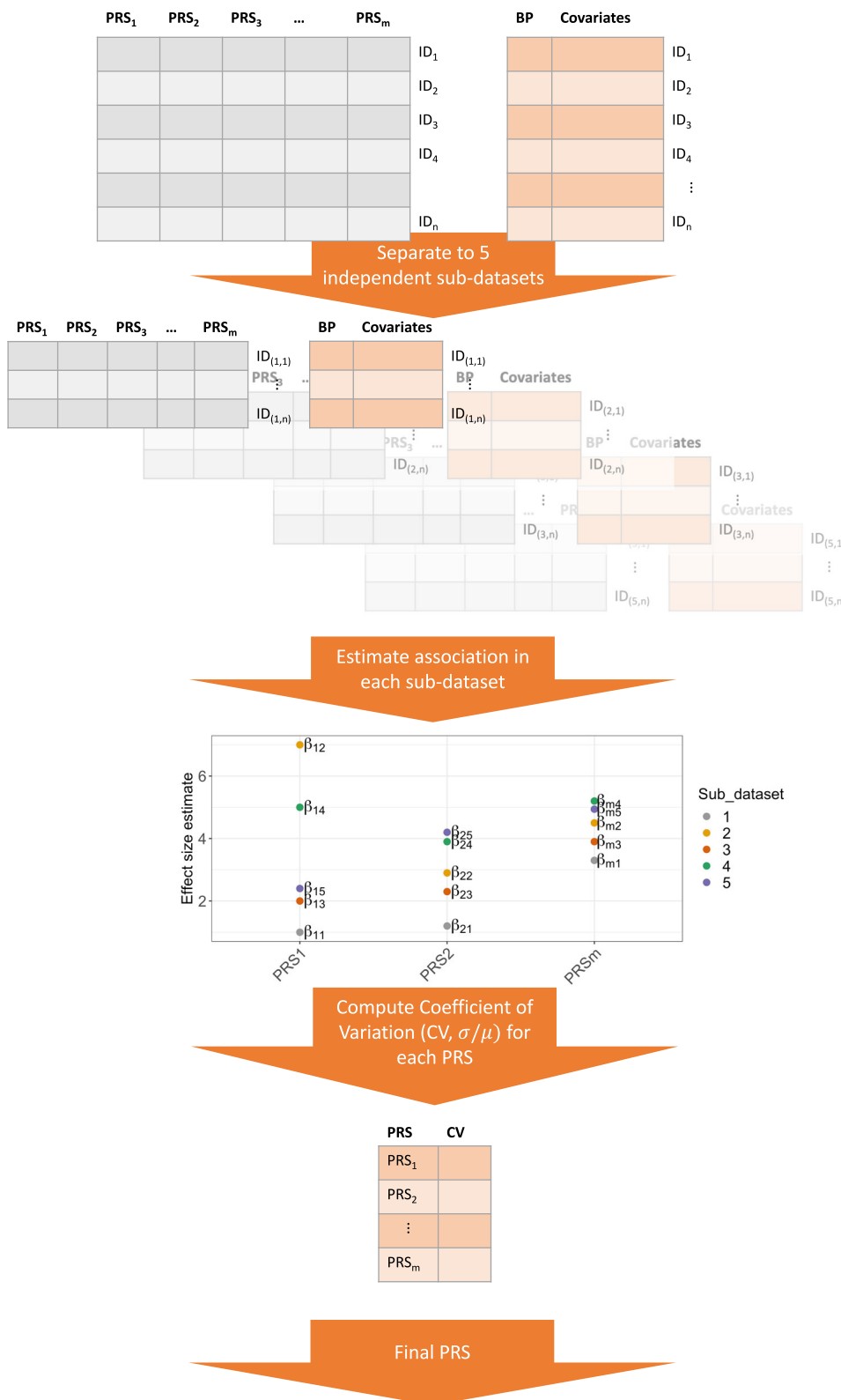

**Fig. 7 PRS selection using coefficient of variation workflow.** Flowchart describing the selection of PRS according to the coefficient of variation (CV) criterion. The data set is split into five independent sub-datasets (without related individuals between the subsets). An association model is fit on each sub-dataset for each PRS. Each PRS, defined by a unique combination of tuning parameter, has 5 independent effect size estimates. We compute the CV for each such PRS, and select the PRS that minimizes the CV.

the test dataset with prevalent hypertension. We used it for follow up analysis of longitudinal measures of hypertension, and in stage 3 and 4 analyses (Fig. 1).

In addition to the PRSsum approach, we also attempted to construct PRS based on methods that account for pleiotropy by modeling the genetic association between traits. Generally, based on the Multi-Trait Analysis of GWAS (MTAG;[47]) framework and SMTPred approach[36], we computed PRS based on (1) computations of new estimated SNP effect sizes genome-wide, by accounting for pleiotropy in each SNP individually, and (2) computations of new PRS as a weighted sum of the trait-specific PRS, here too by accounting for pleiotropy. More information is provided in Supplementary Note 1.

**Association analysis of PRS with hypertension in stage 1 and 2 datasets.** We used logistic mixed models, as implemented in the GENESIS R package[48] version 2.16.1 to estimate the association between the PRS and hypertension, with relatedness modeled via a sparse kinship matrix. Association analyses were adjusted for age, age$^2$, BMI, smoking status (current smoker versus former or never smoker) at the baseline exam, the first 11 PCs, race/ethnicity when evaluating PRS association in a multi-ethnic sample, and time between exams when studying incident associations. We performed two analyses of incident, new onset hypertension in the follow-up exam: one based on individuals who had normal BP at baseline, and second based on individuals who had elevated BP at baseline. We estimated both multi-ethnic and background-specific PRS associations. For the latter, we tested for heterogeneity of estimated effects by race/ethnic background using the Cochran Q test that accounts for covariance between effect estimates across the background groups[49]. Our primary analysis scaled the PRS in all background groups using the same SD, estimated across all TOPMed datasets individuals, for comparability of effect size estimates. In secondary analysis we report effect size estimates per group-specific SDs. We evaluated the predictive performance of PRS by calculating the area under the receiver operating curve (AUC) using the AUC function from the pROC R package[50], version 1.16.2. We used only unrelated individual when calculating AUC. We visualized the unadjusted association of the final HTN-PRS with longitudinal measure of hypertension via a decile plot, demonstrating proportions of individuals in categories of longitudinal BP trajectories in each of the PRS deciles, and assessed the strength of association via linear regression. In secondary analysis, to benchmark the effect of the HTN-PRS against known hypertension risk factors, we compared standardized effect size estimates of the HTN-PRS, BMI, and smoking status, from both the prevalent and incident hypertension analyses.

**Development of new onset hypertension in young adulthood by PRS levels.** Stage 3 dataset consisted of $n = 1388$ self-identified Black and $n = 1699$ self-identified White young adults from CARDIA. Follow up started on average at age 25 (minimum age of participants at baseline = 17). We used 15 years of follow up available on the dbGaP repository[51]. We generated the HTN-PRS for each of CARDIA participant, removed related individuals (degree 3 or higher) and assigned individuals to strata defined by <10 percentile of the PRS, 10–50, 50–90, and >90 percentiles. We first computed these strata across all CARDIA individuals. Next, because there was only a single Black individual in the bottom stratum and only a single White individual in the top stratum, we also defined strata within Black and White groups separately. Next, we fit generalized linear mixed models (GLMM) with random intercept for each participant within strata in the combined and background-specific analyses. The outcome was hypertension, defined as before, and the exposure variables were sex, 11 PCs, age, and age squared (PRS values were not included as explanatory variables). We subtracted the minimum age in the sample, 17, from all age values, for ease of computation of effect later on. The effect estimates of age and age squared are denoted by $\hat{\beta}_{age}$ and $\hat{\beta}_{age^2}$. We used the model to estimate the odds ratio (OR) for hypertension by age relative to the minimum age in the sample, based on the coefficients from the GLMM, i.e., $[(age - 17) \times \hat{\beta}_{age} + (age - 17)^2 \times \hat{\beta}_{age^2}]$ and computed a 95% confidence interval separately for each age, by computing standard errors (SEs) for the above equation based on the estimated SEs and covariances of $\hat{\beta}_{age}$ and $\hat{\beta}_{age^2}$, and assuming multivariate normal distribution of the effect estimates.

**Association of the HTN-PRS with outcomes in the MGB Biobank.** We tested the association of the HTN-PRS with hypertension (another form of replication), coronary artery disease (CAD), ischemic stroke, type 2 diabetes, and chronic kidney disease (CKD), in the MGB Biobank (stage 4 dataset). We also tested the association of the HTN-PRS with obesity because the pan-UKBB GWAS summary statistics that we used for constructing one of the PRS used by the HTN-PRS was from an analysis not adjusted to BMI. We used $n = 40,201$ unrelated individuals with relevant phenotypes. For all outcomes other than CKD we used "curated disease populations" defined by a phenotyping algorithm that uses ICD-9 codes and natural language processing[52]. For CKD we used a single term referring to having a health care system encounter due to CKD ("reason to visit" is CKD). We used logistic regression adjusted for 10 PCs, current age, sex, race/ethnicity. The main analysis was multi-ethnic, and we also tested race/ethnic background-specific associations, though sample sizes were small in non-EA groups. Comprehensive description of the MGB Biobank methods is provided in Supplementary Note 3.

**Secondary analysis of the HTN-PRS using genetic ancestry.** Given fixed variant effect sizes, the distributions of PRS are determined by the frequencies of the alleles used in PRS construction. In admixed populations such as HAs and AAs, and to lower extent, other self-reported race categories, the proportions of genetic ancestries across individuals vary, so that estimated allele frequencies, as well as PRS distributions, may vary depending on the set of individuals used in the analysis. Therefore, we also described the distributions of the within groups defined by high proportions (80% or higher) of a specific genetic ancestry, as well as PRS associations in stage 2 datasets according to these groups. Information about genetic ancestry inference in TOPMed is provided in Supplementary Note 2.

Throughout, all statistical tests are two-sided and are based on normal distribution. We used chi-squared test statistics, a sum of squared normally distributed variables, with one degree of freedom when testing association effect estimates, and chi-squared test statistics with k-1 degrees of freedom when testing heterogeneity of estimated association effects across k strata.

**Reporting summary.** Further information on research design is available in the Nature Research Reporting Summary linked to this article.

## Data availability

TOPMed freeze 8 Whole Genome Sequencing (WGS) data are available under restricted access by application to dbGaP according to the study specific accessions: ARIC: "phs001416", BioMe: "phs001644", CARDIA: "phs001612", CHS: "phs001368", FHS: "phs000974", GENOA: "phs001345", HCHS/SOL: "phs001395", JHS: "phs000964", MESA: "phs001211", WHI: "phs001237". Study phenotypes are available from dbGaP from study accession: ARIC: "phs000090", BioMe: "phs001644", CARDIA: "phs000285", CHS: "phs000287", FHS: "phs000007", GENOA: "phs000379", HCHS/SOL: "phs000810", JHS: "phs000286", MESA: "phs000209", WHI: "phs000200". The Summary statistics from the MVP BP GWAS are available from dbGaP by application to study accession "phs001672". The summary statistics from the PAN-UKBB BP GWAS are available at https://pan.ukbb.broadinstitute.org. MGB Biobank genotyping and phenotypic data are available to Mass General Brigham investigators with required approval from the Mass General Brigham Institutional Review board (IRB). The data to construct the HTN-PRS generated in this study are available in the GitHub repository https://github.com/nkurniansyah/Hypertension_PRS. Source Data displayed in Figs. 2–6 are provided with this paper.

## Code availability

We provide developed scripts used to perform analyses described in the paper, code to construct the HTN-PRS, and code for using the Source Data to generate Figs. 2–6, in the GitHub repository https://github.com/nkurniansyah/Hypertension_PRS.

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

## Acknowledgements

We gratefully acknowledge the studies and participants who provided biological samples and data for TOPMed and CCDG. TOPMed and CCDG acknowledgements, as well as descriptions, acknowledgements, and ethics statements of contributing studies are provided in Supplementary Notes 3 and 4. The views expressed in this manuscript are those of the authors and do not necessarily represent the views of the National Heart, Lung, and Blood Institute; the National Institutes of Health; or the U.S. Department of Health and Human Services. T.S., N.K., and S.R. were supported by National Heart Lung and Blood Institute grants R35HL135818 (S.R.) and R21HL145425 (T.S.). N.F. is supported by NIH DK117445 (N.F.) and MD012765 (N.F.). D.S. was funded by R01 HL117323 (D.S.) from NHLBI/NIH. C.K. was supported by NHLBI grant R01HL151152 (C.K.). TNK received analysis support for this project from NHLBI grant U01HL120393 (B.M.P.).

## Author contributions

T.S. and N.K. conceptualized the study and drafted the manuscript. N.K. performed association analyses. T.N.K., K.L.W., E.V.F., J.H., R.A.J.S., Y.G., N.F., and N.K. prepared and harmonized study-specific phenotype files. T.N.K., J.C.B., X.G., W.P., K.D.T., H.J.L., D.S., J.A.S., S.L., S.W.-S., J.E.M., D.M.L.-J., S.S.R., R.J.F.L., S.R., A.C., C.K., M.F., R.C.K., B.M.P., J.I.R., A.C.M., D.L. designed aspects of the participating cohorts study, including data collections, M.O.G., T.N.K., T.E., K.L.W., J.C.B., X.G., W.P., K.D.T., H.J.L., J.H., Y.G., D.S., J.A.S., B.Y., E.V.F., R.A.J.S., Z.W., S.-J.H., S.L., S.W.-S., J.E.M., D.M.L.-J., S.S.R., R.J.F.L., S.R., A.C., C.K., M.F., R.C.K., B.M.P., J.I.R., D.K.A., A.C.M., N.F., and D.L. critically reviewed and edited the manuscript. All authors reviewed the final version of the manuscript.

## Competing interests

B.M.P. serves on the Steering Committee of the Yale Open Data Access Project funded by Johnson & Johnson. All other co-authors declare no competing interests.

## Additional information

[1]Division of Sleep and Circadian Disorders, Brigham and Women's Hospital, Boston, MA, USA. [2]Department of Medicine, Brigham and Women's Hospital, Harvard Medical School, Boston, MA, USA. [3]Department of Epidemiology, Tulane University School of Public Health and Tropical Medicine, New Orleans, LA, USA. [4]Department of Medicine, University of Miami Miller School of Medicine, Miami, FL, USA. [5]Cardiovascular Health Research Unit, Department of Medicine, University of Washington, Seattle, WA, USA. [6]The Institute for Translational Genomics and Population Sciences, Department of Pediatrics, The Lundquist Institute for Biomedical Innovation at Harbor-UCLA Medical Center, Torrance, CA, USA. [7]Department of Medicine, Columbia University Medical Center, New York, NY, USA. [8]Division of Public Health Sciences, Fred Hutchinson Cancer Center, Seattle, WA, USA. [9]The Jackson Heart Study, University of Mississippi Medical Center, Jackson, MS, USA. [10]Department of Medicine, Columbia University Irving Medical Center, New York, NY, USA. [11]Department of Epidemiology, University of Michigan School of Public Health, Ann Arbor, MI, USA. [12]Human Genetics Center, Department of Epidemiology, Human Genetics and Environmental Sciences, School of Public Health, University of Texas Health Science Center at Houston, Houston, TX, USA. [13]The Charles Bronfman Institute for Personalized Medicine, Icahn School of Medicine at Mount Sinai, New York, NY, USA. [14]Department of Biostatistics, Boston University, Boston, MA, USA. [15]Center for Global Cardiometabolic Health and Departments of Epidemiology, Medicine, and Surgery, Brown University, Providence, RI, USA. [16]Department of Epidemiology & Population Health, Department of Pediatrics, Albert Einstein College of Medicine, Bronx, NY, USA. [17]Department of Epidemiology, Harvard T.H. Chan School of Public Health, Boston, MA, USA. [18]Department of Preventive Medicine, Northwestern University, Chicago, IL, USA. [19]Center for Public Health Genomics, University of Virginia School of Medicine, Charlottesville, VA, USA. [20]Departments of Medicine and Pediatrics, University of Mississippi Medical Center, Jackson, MS, USA. [21]Brown Foundation Institute of Molecular Medicine, McGovern Medical School, University of Texas Health Science Center at Houston, Houston, TX, USA. [22]Department of Epidemiology and Population Health, Albert Einstein College of Medicine, Bronx, NY, USA. [23]Cardiovascular Health Research Unit, Departments of Medicine, Epidemiology, and Health Systems and Population Health, University of Washington, Seattle, WA, USA. [24]College of Public Health, University of Kentucky, Lexington, KY, USA. [25]Department of Epidemiology, University of North Carolina, Chapel Hill, NC, USA. [26]The Population Sciences Branch of the National Heart, Lung and Blood Institute, Bethesda, MD, USA. [27]The Framingham Heart Study, Framingham, MA, USA. [28]Department of Biostatistics, Harvard T.H. Chan School of Public Health, Boston, MA, USA. *A full list of members and their affiliations appears in the Supplementary Information. *A list of authors and their affiliations appears at the end of the paper. ✉email: tsofer@bwh.harvard.edu

## the NHLBI Trans-Omics in Precision Medicine (TOPMed) Consortium

Joshua C. Bis[29], Xiuqing Guo[30], Kent D. Taylor[30], Henry J. Lin[30], Jeffrey Haessler[31], Yan Gao[32], Jennifer A. Smith[33], Simin Liu[34], Sylvia Wassertheil-Smoller[35], JoAnn E. Manson[36,37], Stephen S. Rich[38], Ruth J. F. Loos [13], Susan Redline[36,39], Adolfo Correa[40], Charles Kooperberg[31], Myriam Fornage[41,42], Robert C. Kaplan[31,43], Bruce M. Psaty[44], Jerome I. Rotter[30], Donna K. Arnett[45], Nora Franceschini[46], Daniel Levy[47,48] & Tamar Sofer[36,39,49]

[29]Cardiovascular Health Research Unit, Department of Medicine, University of Washington, Seattle, WA, USA. [30]The Institute for Translational Genomics and Population Sciences, Department of Pediatrics, The Lundquist Institute for Biomedical Innovation at Harbor-UCLA Medical Center, Torrance, CA, USA. [31]Division of Public Health Sciences, Fred Hutchinson Cancer Center, Seattle, WA, USA. [32]The Jackson Heart Study, University of Mississippi Medical Center, Jackson, MS, USA. [33]Department of Epidemiology, University of Michigan School of Public Health, Ann Arbor, MI, USA. [34]Center for Global Cardiometabolic Health and Departments of Epidemiology, Medicine, and Surgery, Brown University, Providence, RI, USA. [35]Department of Epidemiology & Population Health, Department of Pediatrics, Albert Einstein College of Medicine, Bronx, NY, USA. [36]Department of Medicine, Brigham and Women's Hospital, Harvard Medical School, Boston, MA, USA. [37]Department of Epidemiology, Harvard T.H. Chan School of Public Health, Boston, MA, USA. [38]Center for Public Health Genomics, University of Virginia School of Medicine, Charlottesville, VA, USA. [39]Division of Sleep and Circadian Disorders, Brigham and Women's Hospital, Boston, MA, USA. [40]Departments of Medicine and Pediatrics, University of Mississippi Medical Center, Jackson, MS, USA. [41]Human Genetics Center, Department of Epidemiology, Human Genetics and Environmental Sciences, School of Public Health, University of Texas Health Science Center at Houston, Houston, TX, USA. [42]Brown Foundation Institute of Molecular Medicine, McGovern Medical School, University of Texas Health Science Center at Houston, Houston, TX, USA. [43]Department of Epidemiology and Population Health, Albert Einstein College of Medicine, Bronx, NY, USA. [44]Cardiovascular Health Research Unit, Departments of Medicine, Epidemiology, and Health Systems and Population Health, University of Washington, Seattle, WA, USA. [45]College of Public Health, University of Kentucky, Lexington, KY, USA. [46]Department of Epidemiology, University of North Carolina, Chapel Hill, NC, USA. [47]The Population Sciences Branch of the National Heart, Lung and Blood Institute, Bethesda, MD, USA. [48]The Framingham Heart Study, Framingham, MA, USA. [49]Department of Biostatistics, Harvard T.H. Chan School of Public Health, Boston, MA, USA.

