## [Peer Review File · Nature Communications]

A multi-ethnic polygenic risk score is associated with hypertension prevalence and progression throughout adulthoodREVIEWER COMMENTS

Reviewer #1 (Remarks to the Author):

This is a very thorough analysis on the relation of a BP PRS for hypertension prediction in large multi-ethnic cohorts. In addition to clinical value, it also provides a novel approach for developing and constructing predictive PRSs.

I have only a few comments:

1. Abstract sentence: "The HTN-PRS was associated with new onset hypertension among individuals with normal (respectively, elevated) BP at baseline: OR=1.71, 95% CI [1.55, 1.91] (OR=1.48, 95% CI [1.27, 1.71])." is a bit difficult to read/understand.
2. "with perhaps less individuals in the "improved" category in the higher HTN-PRS deciles compared to the lower ones." Please test this with a statistical method so that it would not sound like an opinion/gut feeling.
3. Do the authors have any ideas/hypotheses why black individuals have on average a higher PRS than other races? Is this (or lifestyles) the main reason why black African Americans have a higher rate of hypertension? Please discuss.
4. "Finish"  "Finnish"

Reviewer #2 (Remarks to the Author):

In their manuscript Sofer et al. developed a polygenic risk score for hypertension (HTN) using pan-ancestry GWAS summary statistics of three highly correlated traits (systolic blood pressure [SBP], diastolic blood pressure [DBP], and HTN). They applied a multistage approach to avoid overfitting, combined PRS of each trait via tuning steps and an unweighted summation, and explored the final PRS's performance across various self-reported ethnic groups and for longitudinal data on HTN.

The work is presented in a transparent fashion and covers an important topic; however, I do have several concerns that when addressed might improve the relevance and quality of the presented research:

1. Several previous studies already showed the benefit of multi-trait predictors that combined GWAS results from multiple traits, especially in the presence of pleiotropy or highly / genetically correlated traits, which is obviously the cases for the analyzed traits (SBP, DBP and HTN) (e.g., Maier RM, et al. Improving genetic prediction by leveraging genetic correlations among human diseases and traits. *Nat Commun.* 2018. PMID: 29515099; Li C, et al. Improving genetic risk prediction by leveraging pleiotropy. *Hum Genet.* 2014. PMID: 24337655; Maier R, et al. Joint analysis of psychiatric disorders increases accuracy of risk prediction for schizophrenia, bipolar disorder, and major depressive disorder. *Am J Hum Genet.* 2015. PMID: 25640677).

Furthermore, a recent study proposed a unified framework for a similar setup that also uses summary statistics to improve cross-population / multi-ethnic trait prediction (Cai M, et al. A unified framework for cross-population trait prediction by leveraging the genetic correlation of polygenic traits. *Am J Hum Genet.* 2021. PMID: 33770506.)

So, while the current study excels though their access to the diverse TopMed and MGB data and valuable longitudinal data, there seem to be a missed opportunity to apply (or at least compare the novel method with) existing methods that consider pleiotropy in their PRS construction. By simply

adding the scaled PRS, the resulting PRS might be biased toward genetic risk variants that are shared between all three traits while uniquely contributing effects might have been diluted.

2. The multi-ethnic GWASs that provided the summary statistics for stage 1 seem heavily biased towards European ancestry groups (both > 85% White), especially the "Pan UKBB" data seem to only include a very small fraction of non-White individuals; association signals are thus likely driven by the much larger White populations. Consequently, these GWASs seem not ideal for the development of widely applicable multi-ethnic PRS.

3. Considering the availability of genetic data, inferred ancestry would be a much more appropriate choice for defining subgroups for PRS analyses, especially, when considering the low reliability of self-reported ancestry (Burnett MS, et al. Reliability of self-reported ancestry among siblings: implications for genetic association studies. *Am J Epidemiol.* 2006 PMID: 16421243). Also, the self-reported Asian American (AsA) subgroup likely represents a very heterogeneous group, i.e., East Asian, South East Asian, and South Asian, Pacific Islander, etc..

4. The different PRS distributions by race/ethnic background suggest that the PRS should be scaled within each group for better comparability of effect sizes, e.g., it's unclear if the odds ratios in Figure 6 are per S.D. within the group or within the whole cohort.

Reviewer #1 (Remarks to the Author):

This is a very thorough analysis on the relation of a BP PRS for hypertension prediction in large multi-ethnic cohorts. In addition to clinical value, it also provides a novel approach for developing and constructing predictive PRSs.

Response: Thank you!

I have only a few comments:

1. Abstract sentence: "The HTN-PRS was associated with new onset hypertension among individuals with normal (respectively, elevated) BP at baseline: OR=1.71, 95% CI [1.55, 1.91] (OR=1.48, 95% CI [1.27, 1.71])." is a bit difficult to read/understand.

Response: We agree, and we changed this sentence to:

"At baseline, the HTN-PRS was associated with new onset hypertension among individuals with normal BP (OR=1.71, 95% CI [1.55, 1.91]) and those with elevated BP (OR=1.48, 95% CI [1.27, 1.71])"

2. "with perhaps less individuals in the "improved" category in the higher HTN-PRS deciles compared to the lower ones." Please test this with a statistical method so that it would not sound like an opinion/gut feeling.

Response: This is a great idea, thank you. We now added a regression analysis, where we estimated, separately in each BP category, the association between HTN-PRS decile and the number of people from that category. We first reference this analysis in the methods section in page 13 (new text in bold):

"We visualized the unadjusted association of the final HTN-PRS with longitudinal measure of hypertension via a decile plot, demonstrating proportions of individuals in categories of longitudinal BP trajectories in each of the PRS deciles, and assessed the strength of association via linear regression."

Next, we describe the results in the Results section, "Distributions of longitudinal BP categories across deciles of the HNT-PRS" subsection, in page 18:

"Figure 5 visualizes the distribution of the longitudinal BP categories across deciles of the HTN-PRS, and Table S5 in the Supplementary Information provides results from an analysis using linear regression to test for a linear change in the number of individuals from each BP category as a function of HTN-PRS decile. Indeed, higher deciles have higher proportions of individuals with severe BP category (having hypertension already at baseline) with p-value<0.001 indicating increase in the number of individuals in this category in each decile, and lower deciles have higher proportions of individuals who were free of hypertension in both exams (p-value<0.001 indicating a decrease in the number of individuals in the "always normal or elevated" category with increasing HTN-PRS deciles). Relatively few individuals were categorized as "worsened" or "improved" (transitioning between normal BP, elevated BP, and HTN between exams).

No association was observed with the number of individuals in the “worsened” category in HTN-PRS deciles (p-value=0.21), while the number of individuals in the “improved” category decreased with increasing HTN-PRS deciles (p-value<0.001).”

Finally, Table S5 in the Supplementary Information provides the results from this analysis.

3. Do the authors have any ideas/hypotheses why black individuals have on average a higher PRS than other races? Is this (or lifestyles) the main reason why black African Americans have a higher rate of hypertension? Please discuss.

4. "Finish"  "Finnish"

Response: This is now fixed, thank you.

Reviewer #2 (Remarks to the Author):

In their manuscript Sofer et al. developed a polygenic risk score for hypertension (HTN) using pan-ancestry GWAS summary statistics of three highly correlated traits (systolic blood pressure [SBP], diastolic blood pressure [DBP], and HTN). They applied a multistage approach to avoid overfitting, combined PRS of each trait via tuning steps and an unweighted summation, and explored the final PRS’s performance across various self-reported ethnic groups and for longitudinal data on HTN.

The work is presented in a transparent fashion and covers an important topic; however, I do have several concerns that when addressed might improve the relevance and quality of the presented research:

Response: Thank you for the reviewing our paper and for the summary. We appreciate your excellent comments.

1. Several previous studies already showed the benefit of multi-trait predictors that combined GWAS results from multiple traits, especially in the presence of pleiotropy or highly / genetically correlated traits, which is obviously the cases for the analyzed traits (SBP, DBP and HTN) (e.g., Maier RM, et al. Improving genetic prediction by leveraging genetic correlations among human diseases and traits. *Nat Commun.* 2018. PMID: 29515099; Li C, et al. Improving genetic risk prediction by leveraging pleiotropy. *Hum Genet.* 2014. PMID: 24337655; Maier R, et al. Joint analysis of psychiatric disorders increases accuracy of risk prediction for schizophrenia, bipolar disorder, and major depressive disorder. *Am J Hum Genet.* 2015. PMID: 25640677). Furthermore, a recent study proposed a unified framework for a similar setup that also uses summary statistics to improve cross population/ multi-ethnic trait prediction (Cai M, et al. A unified framework for cross-population trait prediction by leveraging the genetic correlation of polygenic traits. *Am J Hum Genet.* 2021. PMID: 33770506.)

So, while the current study excels though their access to the diverse TopMed and MGB data and valuable longitudinal data, there seem to be a missed opportunity to apply (or at least

compare the novel method with) existing methods that consider pleiotropy in their PRS construction. By simply adding the scaled PRS, the resulting PRS might be biased toward genetic risk variants that are shared between all three traits while uniquely contributing effects might have been diluted.

Response: Thank you for appropriately challenging us to perform an analysis constructing new PRS in a way that explicitly model pleiotropy. This was absolutely the right thing to do and we were ready to change HTN-PRS based on this analysis if the results indicate that a different PRS is better. At the end, these PRS did not perform better than that unweighted sum of the scaled PRS, and we think that this is due to the genetic diversity of the study population from the GWAS. We reference this analysis in the manuscript in the following sections:

In the methods section, page 12, subsection “PRS construction based on multiple GWAS” we added the paragraph:

“In addition to the PRSsum approach, we also attempted to construct PRS based on methods that account for pleiotropy by modelling the genetic association between traits. Generally, based on the Multi-Trait Analysis of GWAS (MTAG; (36)) framework and SMTPred approach (37), we computed PRS based on (1) computations of new estimated SNP effect sizes genome-wide, by accounting for pleiotropy in each SNP individually, and (2) computations of new PRS as a weighted sum of the trait-specific PRS, here too by accounting for pleiotropy. More information is provided in the Supplementary Information.”

Next, in the results section, top of page 18, subsection “PRS associations with baseline hypertension in the stage 2 dataset”:

“In secondary analysis, we compared the HTN-PRS to four additional PRS constructed using approaches that specifically model pleiotropy between the BP traits. Results are provided in Figures S11 (stage 1 dataset) and S12 (stage 2 dataset) in the Supplementary Information. The HTN-PRS had better performance.”

And in the discussion section, page 25:

“While PRSsum is an unweighted sum of PRS, a weighted sum (or with adaptive weights) can be constructed as well (46,47). In particular, there are published methods for leveraging pleiotropy between multiple traits for both discovery of genetic associations and for creating potentially more powerful PRS for each of the traits. We attempted to implement a few of these methods, based on two models of multivariate associations between traits that account for pleiotropy (37,48). However, PRSsum without adaptive weights had stronger associations with hypertension in both stage 1 and stage 2 datasets. We think that PRSsum was more robust because the other methods had to rely on LD inference (for estimating heritability and genetic correlations from summary statistics (49), for generating LD scores, and for genetic SBLUPs (50)). For LD inference we used our own TOPMed datasets, because we do not

have access to the multi-ethnic data that were used for generating the GWAS summary statistics, and moreover, the datasets used to generate the SBP and DBP GWAS are different than the dataset used for generating the hypertension GWAS. Future work should study weighted combination of PRS in diverse populations, including where the PRS were developed based on GWAS in diverse populations.”

We provided methods and results in the Supplementary Materials file (pages 3-4). We do not copy the text here because of the length of the text. However, note that we provide the estimated heritabilities and genetic correlations using LD-score regression in Supplementary Figure S10, and figures comparing the simple PRSsum (the HTN-PRS) to the pleiotropy model-based PRS in stage 1 and stage 2 datasets in Figures S11 and S12.

2. The multi-ethnic GWASs that provided the summary statistics for stage 1 seem heavily biased towards European ancestry groups (both > 85% White), especially the “Pan UKBB” data seem to only include a very small fraction of non-White individuals; association signals are thus likely driven by the much larger White populations. Consequently, these GWASs seem not ideal for the development of widely applicable multi-ethnic PRS.

Response: The “Pan UKBB” is indeed heavily White (and in fact we corrected the fraction listed in Table 1 and it is a bit higher than was listed previously). However, MVP has 69.1% White individuals. While we agree that these GWAS may not be ideal, we think that there are no better GWAS, and that an “ideal” GWAS does not exist right now, unfortunately. To address your comment, we added this sentence to the limitation section of the discussion (one before last paragraph; bottom of page 26):

“While the GWAS that we used (MVP and Pan-UKBB) are the largest ones available with multi-ethnic populations, individuals of European ancestry individuals are still over-represented in these GWAS: 61.9% in MVP and 93% in Pan-UKBB. Future work should utilize additional, diverse, sources of summary statistics as they become available.”

3. Considering the availability of genetic data, inferred ancestry would be a much more appropriate choice for defining subgroups for PRS analyses, especially, when considering the low reliability of self-reported ancestry (Burnett MS, et al. Reliability of self-reported ancestry among siblings: implications for genetic association studies. Am J Epidemiol. 2006 PMID: 16421243). Also, the self-reported Asian American (AsA) subgroup likely represents a very heterogenous group, i.e., East Asian, South East Asian, and South Asian, Pacific Islander, etc..

Response: This is a complex issue that we have spent a long time thinking about. Our perspective is guided by our extensive experience working with admixed populations, and primarily with Hispanics/Latinos. Admixed populations have more than one genetic ancestry.

Hispanics/Latinos are 3-way admixed with European,

Amerindian, and (west) African ancestries. Most Hispanics/Latinos in, e.g. the Hispanic Community Health Study/Study of Latinos (HCHS/SOL), **could not be assigned to a group based on inferred ancestry**. For example, an individual may have, say, 50% European ancestry, 40% Amerindian ancestry, and 10% African ancestry. This figure is, taken from the Conomos et al. "Genetic diversity and association studies in US Hispanic/Latino populations: applications in the Hispanic Community Health Study/Study of Latinos." (2016), demonstrates it. In fact, the groupings used in this figure (Dominican, Puerto Rican, etc.) show similarity in the larger PC space, even though based on proportion ancestry, one can see that there are individuals with both high and low proportions of European ancestry (for example) in most groups. While we agree with the reviewer that the use of self-reported race/ethnicity variables is not ideal, these variables are not biological, and their correlation with genetic ancestry changes with time based on social norms related to the use of these variables, we think that it is appropriate to *assess* the application of PRS within such groups. Self-identification by race and ethnicity is still done in various settings in the U.S. and therefore the *assessment* of biomarkers and their use within these categories is appropriate. This is different than *developing* a genetic measure that would be specific to a specific social category. In fact, while not exactly comparable, NIH has a policy that in phase 3 clinical trials, analyses should be done by strata of gender and of race and ethnicity (<https://nexus.od.nih.gov/all/2022/03/10/designing-analyses-by-sex-or-gender-race-and-ethnicity-in-nih-defined-phase-3-clinical-trials/>), supporting the notion that it is important to assess a biomarker (which may lead one day for an intervention) within race and ethnicity strata.

Therefore, to address your comment, we added a secondary analysis by genetic ancestry. We now added a subsection to the Methods section (bottom of page 15) titled **"Secondary analysis of the HTN-PRS using genetic ancestry"**:

"Given fixed variant effect sizes, the distributions of PRS are determined by the frequencies of the alleles used in PRS construction. In admixed populations such as HAs and AAs, and to lower extent, other self-reported race categories, the proportions of genetic ancestries across individuals vary, so that estimated allele frequencies, as well as PRS distributions, may vary depending on the set of individuals used in the analysis. Therefore, we also described the distributions of the within groups defined by high proportions (80% or higher) of a specific genetic ancestry, as well as PRS associations in stage 2 datasets according to these groups. Information about genetic ancestry inference in TOPMed is provided in the Supplementary Information."

We refer to this analysis in the Results section, subsection (page 22) titled **"Secondary analysis of the HTN-PRS using genetic ancestry"**:

"In an additional secondary analysis, we created groups of individuals defined by having at least 80% of a specific genetic ancestry: at baseline, 5,447 individuals with at least 80% African ancestry, and similarly 97, 783, and 20,939 individuals with at least 80% Amerindian, East Asian, and European ancestry. Sample sizes are smaller when excluding individuals with hypertension at baseline. Notably, most of the HA individuals could not participate in this analysis because they do not have a single predominant ancestry. The distributions of the PRS in each of these groups

(supplemental Figure S14) show that PRS distributions differ between ancestries, due to differences in allele frequencies between them. Figure S15 further provides results from association analysis with hypertension at baseline and incident hypertension in the stage 2 dataset. While the effect size of the PRS per 1 SD of the PRS (with SD computed over all the TOPMed dataset) is largest in the European ancestry group, at the baseline hypertension analysis, the AUC is about the same in the African (0.76) and European ancestry (0.75) groups.”

We were cautious to not over-interpret these results as the sample sizes are small.

We added a sentence in the second paragraph of the discussion, to the part where we mentioned PRS distributions (bottom of page 23, new text in bold):

“The distribution of the various constructed PRS, including the final HTN-PRS (PRSsum based on Selected CV-PRS), differed across race/ethnic backgrounds. This is expected, because PRS are sums of alleles, which have different distributions (defined by allele frequencies) across genetic ancestries, and therefore, also race/ethnic background, as these generally have different ancestry admixture. **Indeed, PRS distributions also differed when assessed over groups constructed using individuals with high proportions of specific genetic ancestries.**”

Another two points:

(a) Asian Americans in the stage 2 dataset are not very heterogeneous. They are all of Chinese descent and are from the MESA study.

(b) Genetic ancestry, while genetically is considerably a more precise term compared to race and ethnicity, is also not 100% precise. Genetic ancestries may be defined and inferred in various ways (e.g. European ancestry based on Whites individuals from various origins, or more fine-grained: British European ancestry, etc. For example, see this preprint <https://www.biorxiv.org/content/10.1101/2022.02.17.480829v1.abstract> about fine population structure and imputation of individuals from French, suggesting that, given appropriate data, one may be able to distinguish French ancestry from other specific European ancestries).

4. The different PRS distributions by race/ethnic background suggest that the PRS should be scaled within each group for better comparability of effect sizes, e.g., it's unclear if the odds ratios in Figure 6 are per S.D. within the group or within the whole cohort.

Response: We agree that it is important to make the effect sizes comparable. We scaled the PRS in the same manner in each group in which association analysis was performed: by the SD computed within the whole TOPMed cohort. This is written in section “PRS construction based on a single GWAS”, as follows: “To standardize PRS while keeping effect sizes comparable in all analyses, we computed the mean and standard deviations (SDs) of each type of PRS based on the complete TOPMed dataset. Then, we used these means and SDs in all subsequent analyses: for a given PRS in any dataset, we subtracted its pre-computed mean and divided it by its pre-

computed SD. This standardization approach allowed for obtaining comparable effect sizes across stage 1, 2, 3, and 4 datasets, as well as across background-specific and multi-ethnic analyses. Standardization does not influence p-values or prediction measures.”

We understand that readers may miss this information due to where it is placed in the text. Therefore, we edited the methods section “Association analysis of PRS with hypertension in stage 1 and 2 datasets”, page 13, to include the sentence:

“Our primary analysis scaled the PRS in all background groups using the same SD, estimated across all TOPMed datasets individuals, for comparability of effect size estimates. In secondary analysis we report effect size estimates per group-specific SDs.”

We also added this information in the legends of figures 3 and 6, using the additional sentence:

“PRS SD was defined according to the sampling SD of the PRS estimated in the entire TOPMed dataset.”

Because we also understand that readers may be interested in understanding differences in effect sizes by group-specific scaling, we also added supplemental figure S6, which mimics figure 6, but now with effect sizes provided based on within-group scaling of the PRS.

Finally, we reference this analysis in the results section “HTN-PRS association with prevalent and incident hypertension across race/ethnic backgrounds”, by adding the sentence:

“Supplementary Figure S6 reports an analysis mimicking that in Figure 6, with now effect sizes reported per 1 SD increase in the PRS where the SD is computed within the group, rather than in all TOPMed. Within European and Asian Americans, the ORs per SD become lower when accounting for group-specific PRS distribution.”

REVIEWERS' COMMENTS

Reviewer #1 (Remarks to the Author):

The authors have adequately answered my questions/comments although I do not see a response for comment #3. I was just expecting a sentence or two of speculation on whether these genetic differences between races could explain some of the observed between-race differences in hypertension prevalence in the US.

Reviewer #2 (Remarks to the Author):

Sofer et al. addressed all my comments with attention to detail in a well-structured response and added valuable information to the manuscript. By doing so they further improved an already excellent research study that will be impactful and relevant for a broad readership. I have no further comments.

Response to review of NCOMMS-21-49580A “A multi-ethnic polygenic risk score is associated with hypertension prevalence and progression throughout adulthood”

We would like to apologize to the editor and reviewer 1 for erroneously skipping over reviewer 1’s comment 3 from the original review of the manuscript. It is an important question, and we address it as follows.

Current reviewer 1 comment:

The authors have adequately answered my questions/comments although I do not see a response for comment #3. I was just expecting a sentence or two of speculation on whether these genetic differences between races could explain some of the observed between-race differences in hypertension prevalence in the US.

Original comment #3:

3. Do the authors have any ideas/hypotheses why black individuals have on average a higher PRS than other races? Is this (or lifestyles) the main reason why black African Americans have a higher rate of hypertension? Please discuss.

Response: These are very good questions (and again, we apologize for skipping them before). It is difficult to speculate why Black individuals have higher PRS values on average. Technically, this is because genetic variants with higher estimated effect size (higher weight) have higher frequency in African ancestries. However, this mathematical fact does not inform of an underlying reason, e.g. whether any population selection process lead to differences in allele frequency in the specific observed pattern, and why. The second part of this question refers to the implication of Black American individuals having higher PRS values: does this drive their higher rate of hypertension? Here too we think it is too early to reply with certainty given yet understudied gene-environment interaction effects. We added the following paragraph to the discussion (page 24, bottom):

“It is notable that AA individuals had higher HTN-PRS values on average, compared to other backgrounds. While, as noted, these distributional differences stem from allele frequency differences, there are two questions that may be asked: First, is there any reason for these difference, such as population-level selection pressure, or are they random? Second, do these differences drive higher rates of hypertension in AA individuals? While our work cannot inform the answer to the first question, the analysis in CARDIA (Figure 7) does suggest that higher PRS values are associated with earlier hypertension, supporting a genetic role for earlier hypertension in AAs. However, this should be interpreted with caution. BP has been consistently shown to be affected by lifestyle, including, lifestyle interacts with genetic variants to increase their estimated effect on BP. It is plausible that detected BP variants tend to be those that interact with dietary and other lifestyle exposures that are more common in individuals with lower socioeconomic status, such as Black Americans. In this case

there could be a dual bias: bias of higher likelihood to detect specific variants that interact with such exposures, and a bias of some background groups, here AA individuals, having higher rates of the same exposures. Therefore, we cannot separate genetic factors from the dietary, lifestyle and environmental factors as their interactions are the ultimate driver determining group differences in hypertension rates.”